# Autonomous and informed decision-making: The case of colorectal cancer screening

Linda N. Douma [1,2]*, Ellen Uiters[2], Marcel F. Verweij[3], Danielle R. M. Timmermans[1,2]

**1** Amsterdam UMC, Vrije Universiteit Amsterdam, Department of Public and Occupational Health, Amsterdam Public Health research institute, Amsterdam, The Netherlands, **2** National Institute for Public Health and the Environment (RIVM), Bilthoven, The Netherlands, **3** Department of Social Sciences, Wageningen University, Wageningen, The Netherlands

* l.douma@amsterdamumc.nl

## Abstract

### Introduction

It is increasingly considered important that people make an autonomous and informed decision concerning colorectal cancer (CRC) screening. However, the realisation of autonomy within the concept of informed decision-making might be interpreted too narrowly. Additionally, relatively little is known about what the eligible population believes to be a 'good' screening decision. Therefore, we aimed to explore how the concepts of autonomous and informed decision-making relate to how the eligible CRC screening population makes their decision and when they believe to have made a 'good' screening decision.

### Methods

We conducted 27 semi-structured interviews with the eligible CRC screening population (eighteen CRC screening participants and nine non-participants). The general topics discussed concerned how people made their CRC screening decision, how they experienced making this decision and when they considered they had made a 'good' decision.

### Results

Most interviewees viewed a 'good' CRC screening decision as one based on both reasoning and feeling/intuition, and that is made freely. However, many CRC screening non-participants experienced a certain social pressure to participate. All CRC screening non-participants viewed making an informed decision as essential. This appeared to be the case to a lesser extent for CRC screening participants. For most, experiences and values were involved in their decision-making.

### Conclusion

Our sample of the eligible CRC screening population viewed aspects related to the concepts of autonomous and informed decision-making as important for making a 'good' CRC screening decision. However, in particular the existence of a social norm may be affecting a true autonomous decision-making process. Additionally, the present concept of informed

**Data Availability Statement:** The data supporting the conclusions of our manuscript are included within the manuscript. The full data set cannot be shared publicly as participants did not give consent for their transcripts to be shared in the public

domain. The transcripts contain identifying items, which are therefore sensitive to privacy issues. Participants consented into the study with the understanding that only anonymised quotations would be publicly available. Anonymised data can be made available to qualified researchers by request to the medical ethical committee of the VU University Medical Centre, who can be contacted at metc@vumc.nl.

**Funding:** This research was funded by the Strategic Programme of the National Institute for Public Health and the Environment (RIVM). The funders had no role in study design, data collection and analysis, decision to publish, or preparation of the manuscript.

**Competing interests:** The authors have declared that no competing interests exist.

**Abbreviations:** CRC, Colorectal cancer screening; ISCED, International Standard Classification of Education; ISO, International Organisation for Standardization.

decision-making with its strong emphasis on making a fully informed and well-considered decision does not appear to be entirely reflective of the process in practice. More efforts could be made to attune to the diverse values and factors that are involved in deciding about CRC screening participation.

# 1. Introduction

## 1.1. Informed decision-making

In Western society today, a strong emphasis is being put on the personal pursuit of freedom, wealth and happiness [1]. This comes with an individualistic perspective and the prominence of personal responsibility, which is also reflected in the medical and public health domain, where active involvement regarding your own health and participating in health care decisions has steadily and increasingly been incorporated [1, 2]. Concepts such as empowerment [3], self-management [4], shared decision-making [5], informed decision-making [6] and autonomy [7] have become an essential part of medical care and public health. This is also evident concerning colorectal cancer (CRC) screening and other types of cancer screening, with many experts in the field of cancer screening currently and increasingly considering it important that people make a personal and informed decision regarding screening participation [8–10]. Consequently, making an informed decision, according to the expert-defined concept of informed decision-making, is generally viewed by experts as making a 'good' screening decision [9–11]. Related to the introduction of informed decision-making in CRC screening, and other types of cancer screening, is the increased awareness that CRC screening involves possible benefits, such as reducing the incidence and mortality of CRC [12–15], but also possible harms and downsides, such as false-negatives, false-positives, overdiagnosis, overtreatment, and risks associated with colonoscopy [16–20]. These benefits, harms and downside are, in general, similar to the benefits, harms and downsides associated with other types of cancer screening, although of course different screening tests are being used. Especially on an individual level, it is not apparent whether the possible benefits outweigh the possible harms and downsides of screening as different people could weigh these differently [9, 18, 20]. Therefore, the question of whether the possible benefits for an individual outweigh the possible harms and downsides is associated with the knowledge and understanding of these benefits, harms and downsides as well as with someone's personal situation and values [9, 21, 22].

Generally, an informed screening decision is commonly defined as:

A deliberative decision based on sufficient and relevant knowledge concerning the different choice-options (i.e. fully informed) and consistent with the decision-maker's values, often operationalised as their attitude towards screening and/or their preferences towards a screening test [8, 23, 24]. A central part of an informed decision is also that it is one's personal and free decision (autonomous).

It is important to note that the concept of informed decision-making is not equivalent to the concept of shared decision-making. In informed decision-making, there is usually no health professional involved in the decision-making process, which is the Dutch situation, or a physician merely provides information about screening. In both cases, there is no discussion between a physician and the decision-maker about the screening information and the decision-maker's values and preferences. In a shared decision-making context, a physician and patient jointly discuss and decide about the medical intervention and this could stimulate a broader perspective on the information, values and preferences involved [5, 25]. Thus, making

an informed decision can also be the result of a shared decision-making process, but the *concept of informed decision-making* as such follows an individual decision-making process, without active involvement of a health professional.

## 1.2. CRC screening in the Netherlands

In the Netherlands, a national preventive screening programme exists for colorectal cancer since January 2014. Breast cancer screening and cervical cancer screening were already implemented (in 1990 and 1996, respectively). From the beginning, informed decision-making regarding CRC screening is emphasised as important [26–28]. The Dutch screening programmes are organised by the government, which is similar to a number of other countries, such as the United Kingdom (UK), Australia, Italy and Finland [29, 30], but also not the standard situation worldwide. With organised cancer screening, people are actively invited to participate in cancer screening and the uptake is monitored. However, other countries, such as Germany or the United States of America (USA), rely on opportunistic screening [31, 32], where, in essence, the responsibility lies with the individuals themselves to ask their physician to be screened, and the uptake may not be monitored. Regarding the Dutch CRC screening programme, everyone between the ages of 55 and 76 years old biennially receives an invitation to participate in CRC screening via a self-administered stool test (immunochemical faecal occult blood test: iFOBT), which is payed for by the government. If the stool test gives a positive result, people are referred for a colonoscopy to find out if they actually have (precursors of) colorectal cancer. This follow-up test is covered by their health insurance. In 2015, 73% of those invited for the CRC screening programme decided they wanted to be screened [33], which is relatively high compared to other countries. It is important to note that the eligible population of CRC screening, and other types of cancer screening, concerns healthy individuals who experience no symptoms.

## 1.3. Informed decision-making and the subject of autonomy

The concept of informed decision-making appeals strongly to the notion of people being autonomous and making autonomous decisions [7]. However, the subject of autonomy within informed-decision making seems to have been interpreted relatively narrowly, with an emphasis on people following their own desires and test-preferences and having 'freedom of choice' and 'agency' regarding CRC screening participation [23, 24] (Fig 1).

Freedom of choice/decision means that there is an actual decision to make and people feeling free to decide what they want or prefer [7, 34, 35]. Regarding CRC screening, this has been translated into giving people the decision between participating and not participating in screening as well as the choice between different screening methods. Agency means that people have, or can take, the responsibility as well as the capacity to make a decision [7, 34, 35]. Regarding CRC screening, this has been translated into communicating to people that the decision is theirs to make, and ensuring there is information available that experts find important to enable people to make this decision on their own. However, a broader interpretation of autonomy would also include 'self-constitution', in addition to freedom of choice and agency. Self-constitution encompasses more than people's attitude or test-preferences towards screening. Self-constitution involves autonomy being related to people's identity, individuality and authenticity, [35–39]. This concerns people living by their values, and expressing and further developing their identity through the pursuit of personal goals and the decisions they make. It entails people having the freedom to be involved in the decisions regarding their own life, to accomplish personal goals and to choose how they want to realise these goals (while respecting the rights of others) [7, 34, 35, 38, 39]. Although this might sound somewhat weighty regarding

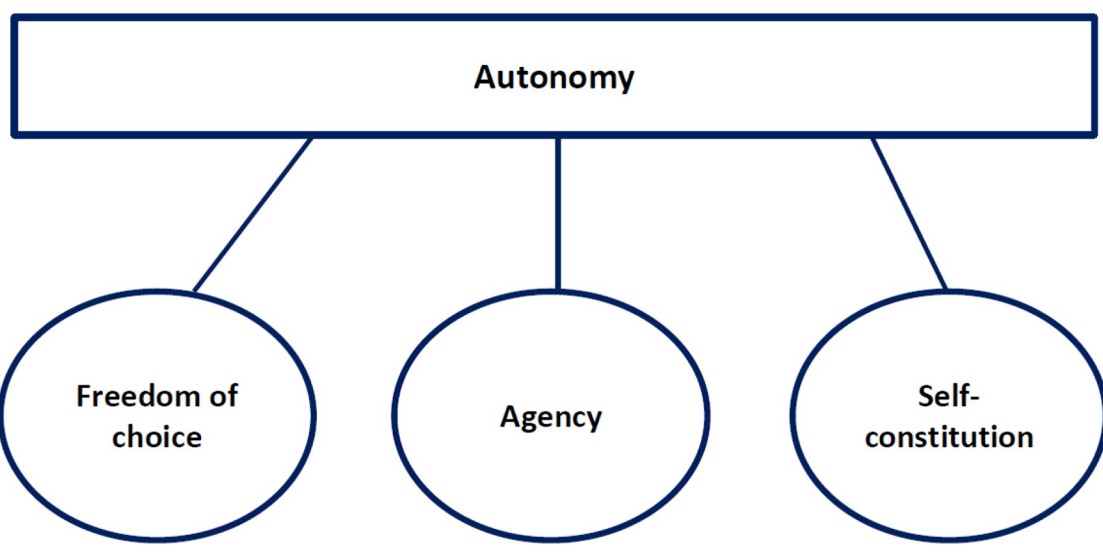

**Fig 1. Conceptual components of autonomy.**

the relatively easy decision concerning CRC screening, we believe it to be of relevance as the values and goals related to this decision also, to a degree, reflect how people value prevention and screening in general. Additionally, the subject of cancer in itself often has a strong affective connotation to many people, thereby appealing to what they find important in life.

## 1.4. Autonomous decision-making in practice

In addition to the subject of autonomy within informed decision-making seemingly being interpreted too narrowly, there might also be concerns regarding the realisation of making an autonomous CRC screening decision in practice. At present, these concerns refer to both the broader and more narrow interpretation of autonomy. Firstly, having freedom of choice should also be experienced as such by the people making the decision. Experiencing the CRC screening decision as an actual and free decision, however, may not always be the case. Jepson et al (2007) [21], for example, found that many people did not view CRC screening as something one could decide to do, but rather as something one is simply expected to do. Additionally, in the case of a government actively offering CRC screening, this could give off the signal that it is a good thing to do, which could lead to people not truly experiencing it as a free decision [22, 40]. Furthermore, actively inviting people to participate in CRC screening means that people are expected to make a decision at that point in time. Thus, they are not entirely free in deciding if, and when, they want to be confronted with this decision. A different concern is that autonomous decision-making also involves having the freedom to decide if one wants to make the decision oneself, or that the involvement of someone else whose judgment is valued is preferred [8]. Several studies found, for example, that many people preferred consultation with their general practitioner or a strong advice from the relevant authorities concerning CRC screening to making the decision themselves [40–42]. Therefore, consciously choosing to rely on the trustworthy opinion of an expert could also be part of autonomous decision-making [41]. Lastly, the concept of informed decision-making could be construed as a form of bio-ethical paternalism, since others (i.e. experts in the field of cancer screening) determine how one should make their screening decision and when this is considered 'good' decision-making [7, 43]. This seems to be in contrast with providing people the freedom to reach their personal goals in a way that they feel is right for them. Thus, the concept of informed decision-making

as it is presently being interpreted and executed, might not be the ideal way for people to achieve autonomy concerning CRC screening participation.

## 1.5. 'Good' decision-making according to the eligible population

Besides the concept of informed decision-making having possible limitations for the making of autonomous CRC screening decisions, it might also not be doing justice to how people make this decision in real life, and what they find important in making a 'good' screening decision [21, 44]. This in comparison to the beforementioned expert view on what constitutes a 'good' screening decision, namely a decision following the concept of informed decision-making [9–11]. Previous studies have found that people regularly failed to make an informed CRC screening decision according to the commonly used expert-definition, because they did not have sufficient knowledge, or their decision did not match their attitude or preferences. However, people themselves did feel informed, were satisfied with their decision and experienced little decisional conflict [45, 46]. Thus, perhaps aspects other than being knowledgeable, such as decision satisfaction or confidence, better reflect what constitutes in real life as being important for making a 'good' screening decision [23, 47, 48]. Of course, it could be that people's decision satisfaction or confidence is associated with being informed [21, 24]. However, it could also be that it is associated with other factors, for example, getting peace of mind [11, 49], having discussed the decision with relevant others [50, 51], or realising their personal goals [36, 37]. Therefore, the question concerns whether making an informed decision according to expert standards should be the main, or sole, focus of what is important in making a 'good' CRC screening decision.

## 1.6. Study aim

Although many studies have been conducted regarding CRC screening and other types of cancer screening and informed decision-making, the majority seems to have focused on examining whether an informed decision had been made [46], and which reasons and factors were associated with screening-uptake or participation [42, 52]. With the exception of a few studies [40, 41], the autonomy aspects of informed decision-making have received relatively little attention. Consequently, relatively little is known about whether the eligible CRC screening population experiences being able to decide freely and according to their own underlying goals and values about CRC screening participation. Additionally, relatively little is known about when the eligible CRC screening population believes to have made a 'good' screening decision. Examining this seems highly relevant as individuals are expected to be personally responsible for their decision concerning CRC screening participation and the possible consequences associated with it. Therefore, using in-depth interviews, which provides us the opportunity to delve in deeper into the process of decision-making, the aim of our study was to explore how the concepts of autonomous and informed decision-making relate to: 1) How the eligible CRC screening population in actuality decide about first-time participation in the Dutch national preventive CRC screening programme; and 2) What the eligible CRC screening population believes to be a 'good' decision concerning CRC screening participation.

## 2. Methods

### 2.1. Ethics approval

The Medical Ethics Review Committee (METC) of VU University Medical Centre has declared that an official approval of this study by their committee is not required (IRB00002991/ FWA00017598). We explained the purpose of our research to respondents, ensured their

anonymity by using codes and obtained their written informed consent before starting the interview.

## 2.2. Recruitment of interview candidates

Interview candidates were recruited via a national online research panel (Flycatcher Internet Research, www.flycatcher.eu; ISO 26362). Members of this panel sign up voluntarily to participate in online research. To thank interviewees for their participation, we provided them with a 20-euro gift card. For our study, we were interested in examining the decision-making process of people who were invited for the first time for CRC screening. This decision-making process would not only happen on the day that people receive their invitation or on the day that they actually take the screening test, but also some time before that. Especially since the existence of the CRC screening programme and who was eligible to be invited had received a reasonable amount of media attention. Therefore, we recruited participants who, based on their birth year, relatively recently received an invitation for CRC screening/made their decision, or who were expected to receive an invitation/make their decision soon. In theory, everyone between 55 and 76 years old was eligible to be invited for the CRC screening programme. However, in the Netherlands, the CRC screening programme was gradually being introduced over a period of five years (2014–2019) among different birth cohorts. According to a publicly available general schedule, the following birth years were supposed to receive an invitation for the first time for CRC screening in 2017 or 2018: 1942, 1943, 1944, 1956, 1958, 1959, 1960, 1961, 1963. In the Netherlands, we do not have a publicly available database of people being invited for CRC screening. Therefore, we used a research panel as they know people's birth year, which we could use as a filter. Thus, eventually, 1233 panel members from the beforementioned birth years were approached via email in April 2017 to participate in our study. However, it was possible that a selection of these people did not actually received an invitation as scheduled as deviations from the invitation schedule occurred. We ended up with 323 respondents who were eligible *and* interested in participating in our study. From these respondents it was also known whether or not they had participated in the CRC screening programme or what their intention was regarding participation. With respondents' permission, the online panel provided the first author (LD) with their contact information, after which a sample of this group was contacted to set up an interview appointment. The interviews were conducted over a period of five months (May-October 2017) and held at interviewees' private homes, their work location or the VU Medical Centre in Amsterdam, according to the interviewees' preference. The aim was to obtain a diverse interview sample with regards to being invited for the CRC screening programme or not, having participated in the CRC screening programme or not, geographic location, sex, age, and education (low, intermediate, high; according to the International Standard Classification of Education (ISCED), 2011). We tried to oversample low educated people as their views can be underrepresented in research. Unfortunately, they were underrepresented in our 'starting-pool' of possible interviewees. Additionally, despite efforts on our end to accommodate them and for diverse reasons (e.g. not picking up their phone, no time, other priorities, other unmentioned experienced barriers), they proved to be more difficult to actually schedule an interview with. Participants were all in the average risk population. We stopped conducting interviews when no new main themes were identified in the data.

## 2.3. Data collection and analysis

We used a qualitative design, using in-depth interviews. LD conducted all interviews, which were semi-structured using open questions. See the *Supporting Information* for the interview guide (S1 File). The interview guide was developed within our research team. We discussed

the main questions relevant to ask in light of the objective of our study, the order of the questions, possible follow-up questions and the wording of the questions. We first conducted two pilot-interviews, after which we made a few small adjustments in the order and wording of the main questions being asked. We asked people about their experience with making, or soon to be making, the CRC screening decision and their views and thoughts concerning the diverse aspects involved in autonomous and informed decision-making. We did not provide them with any exact definitions of, for example, a free or 'good' decision. Our interview guide addressed the following main topics: Why and how did people make their decision about participation in the CRC screening programme; What do they find important in life and how does preventive CRC screening relate to this; What are their experiences and views regarding the decision about participation in the CRC screening programme being a free and voluntary decision of their own; How do they view the role of the government; What was involved in and needed for making their decision about participation in the CRC screening programme; When do they consider to have made a 'good' decision about participation in the CRC screening programme and health-related decision in general. The interview duration ranged from approximately 25 to 55 minutes. With the permission of the interviewees, the interviews were audio recorded. The audio recordings were transcribed verbatim and the transcripts were analysed using Atlas.ti 7.5.17. Transcripts were analysed using descriptive thematic analysis [53, 54], starting with an open coding phase in which the data were examined and themes were categorized. The topics addressed in the interview guide provided a conceptual starting point in this categorization process. LD and the second author (EU) independently coded five of the interviews. Minor differences between the two coders in distinguishing themes and subthemes were discussed and resolved, after which LD analysed the remaining transcripts. The interview guide and the quotes in Table 2 (Results) were translated from Dutch to English by the first author and proofread by a native (UK) English speaker.

## 3. Results

Conducting our qualitative analysis, the following main themes emerged, which were used to structure our results: 1) Aspects related to being informed about CRC screening; 2) Aspects related to autonomous decision-making; 3) Oher aspects involved in the decision-making process; 4) A 'good' screening decision.

### 3.1. Sample characteristics

Our research sample consisted of 27 interviewees: nine (33%) who participated in the CRC screening programme and nine (33%) who had the intention to participate (hereinafter referred to as CRC screening participants); five (19%) who did not participate in the CRC screening programme and four (15%) who had the intention not to participate (hereinafter referred to as CRC screening non-participants). The sample characteristics are summarised in Table 1.

### 3.2. Aspects related to being informed about CRC screening

**3.2.1. Role of information and the official brochure.** Participants and non-participants of the CRC screening programme indicated wanting to know about the basics of more or less the same aspects regarding CRC screening (e.g. the test procedure, the screening purpose, and the possible general benefits, harms and downsides of screening) (Quote 1, Table 2). Most CRC screening participants and non-participants mentioned they heard or were informed about generally all of the aspects they wanted to know about (we did not systematically assess their knowledge). However, some CRC screening participants and non-participants

**Table 1. Characteristics research sample***.

| Variables | Participated in CRC screening / Intention to participate in CRC screening N (%) | Did not participate in CRC screening / Intention not to participate in CRC screening N (%)* | Total sample N (%) |
|---|---|---|---|
| Total | 18 (100) | 9 (100) | 27 (100) |
| *Sex* | | | |
| Male | 10 (56) | 3 (33) | 13 (48) |
| Female | 8 (44) | 6 (67) | 14 (52) |
| *Education* | | | |
| Low | 9 (50) | 1 (11) | 10 (37) |
| Intermediate | 5 (28) | 4 (44) | 9 (33) |
| High | 4 (22) | 4 (44) | 8 (30) |
| *Age category* | | | |
| 55–59 | 7 (39) | 3 (33) | 10 (37) |
| 60–64 | 6 (33) | 2 (22) | 8 (30) |
| 65–69 | 0 (0) | 0 (0) | 0 (0) |
| 70–74 | 5 (28) | 4 (44) | 9 (33) |
| *Relationship status* | | | |
| Currently married/in a relationship | 15 (83) | 7 (78) | 22 (81) |
| Currently not in a relationship | 3 (17) | 2 (22) | 5 (19) |
| *Has children* | | | |
| Yes | 15 (83) | 5 (56) | 20 (74) |
| No | 3 (17) | 4 (44) | 7 (26) |

* Total percentage is in some cases less than 100% due to rounding off

- Because of the gradual introduction of the CRC screening programme in the Netherlands, people from the age category 65–69 were not invited for CRC screening during our study period (2017/2018)

mentioned not to have heard about any downsides or risks associated with CRC screening and were unclear about the target population and procedure of the CRC screening programme. Several CRC screening participants and non-participants mentioned that information was mostly needed regarding what to expect once you have decided to participate in CRC screening. Most CRC screening participants and non-participants had read or browsed through the official brochure accompanying the CRC screening invitation, or intended to do so after receiving the invitation (Quote 2, Table 2). Most already had a clear idea about participating before actually receiving the CRC screening invitation and official brochure (Quote 3, Table 2).

### 3.3. Aspects related to autonomous decision-making

**3.3.1. A free decision and social pressure.** All CRC screening participants and non-participants experienced that they had a choice to either participate or not participate in the CRC screening programme (Quote 4, Table 2). However, many CRC screening non-participants experienced their decision as not being a completely free one (Quote 5, Table 2). They perceived the social norm to be to participate in CRC screening and experienced a certain social pressure–from people close to them as well as from society as a whole. Both CRC screening participants and non-participants indicated that most people in their environment were positive about CRC screening and other types of cancer screening, and were participating in it. A number of CRC screening participants mentioned that cancer screening for women (i.e. the

**Table 2. Illustrative quotes related to the themes of the CRC screening decision-making process.**

| Theme | Illustrative quotes |
|---|---|
| **Role of information and the official brochure** | Quote 1<br>"Of course, the procedure and the usefulness of the screening, that is most important. And the purpose of it and what they hope to achieve with it. That is what you need to know." *(R21, CRC screening non-participant, female, age 72, intermediate education)*<br>Quote 2<br>"It is important that I have access to certain knowledge. And then it is up to me if I want to know all the details or not. Like I said, some things in the brochure I just scanned because before I read it I already thought 'this is a good thing'." *(R19, CRC screening participant, female, age 57, high education)*<br>Quote 3<br>"As soon as I heard about it [CRC screening], I knew I was going to participate. Then I just waited for the invitation letter to come... I have read the brochure, it was clear." *(R5, CRC screening participant, male age 72, high education)* |
| **A free decision and social pressure** | Quote 4<br>"You have the opportunity to participate, but you can also say that you do not want to. And then there is no one telling you 'you should send in your stool-test' or 'why are you not sending the test back?'." *(R15, CRC screening participant, female, age 60, low education)*<br>Quote 5<br>"I do not experience it to be a fully free decision. From the outside, a lot of pressure is being put on you, as in 'just do it, it will not do any harm and it takes you little effort'." *(R22, CRC screening non-participant, female, age 61, high education)*<br>Quote 6<br>"Of course, other people have an opinion about that [my decision not to participate], but that does not make me... well, maybe a little bit... that you start getting some doubt, 'what if I do get colorectal cancer, then they will say "I told you so"'. But right now I would just say, as I am feeling mentally strong, 'tough luck, too bad then'." *(R24, CRC screening non-participant, female, age 61, low education)* |
| **One's own decision and the involvement of others** | Quote 7<br>"I believe everyone should decide that for themselves, if they want to participate or not, and what they think of it." *(R2, CRC screening participant, female, age 61, low education)*<br>Quote 8<br>"If I have to make a decision about my body and myself, then I do not think anyone else should be involved in that." *(R22, CRC screening non-participant, female, age 61, high education)* |
| **Role of the government and making screening participation obligatory** | Quote 9<br>"That the government is behind this [CRC screening], that is fine by me, that they want to encourage people a little bit... but I believe most of it [health responsibility] is up to people themselves." *(R4, CRC screening participant, female, age 60, low education)*<br>Quote 10<br>"The health care facilities have to come about in some way. I do believe that to be a typical government responsibility. But we are talking about prevention now, or perhaps even the promotion of health. Then yes, how far can you go as a government? For me there are limits at some point." *(R26, CRC screening non-participant, male, age 56, high education)*<br>Quote 11<br>"Well, for me, I would not mind if it [CRC screening] was more mandatory. But I think in general people would like to decide this for themselves." *(R9, CRC screening participant, male, age 74, high education)* |
| **Role of life values** | Quote 12<br>"Well, I believe being healthy is most important in life, no millions can compete with that. I know that from experience, so doing this [CRC screening] really resonates with who I am." *(R10, CRC screening participant, male, age 60, intermediate education)*<br>Quote 13<br>"I am not a person who wants to know or prevent everything and is very cautious in not doing this or that. For me it is important that I am enjoying myself and part of that is up to you." *(R21, CRC screening non-participant, female, age 72, intermediate education)*<br>Quote 14<br>"Well, I think, everyone naturally finds their health important. [...] And I think that I find my freedom important, and that this screening examination is a certain threat to that. I believe it will affect my freedom because if I end up in the medical circuit I will not feel free anymore." *(R26, CRC screening non-participant, male, age 56, high education)* |

*(Continued)*

**Table 2.** (Continued)

| Theme | Illustrative quotes |
|---|---|
| **The perception and personal weighing of the benefits, harms and downsides of CRC screening** | Quote 15<br>"Well, early this year I knew I was going to be invited [for CRC screening] and then I started to doubt, 'shall I do it, shall I not do it'. [. . .] But when you read up on it and read about what the risks are, I thought 'this is not for me'." *(R27, CRC screening non-participant, female, age 72, intermediate education)*<br>Quote 16<br>"There is a certain risk that you will be confronted with a false-positive result. There can be many causes for blood in your stool, it does not have to be a sign of colorectal cancer." *(R26, CRC screening non-participant, male, age 56, high education)*<br>Quote 17<br>"If they find something [as a result of CRC screening] then that is a downside, but I accept that as something that comes with it." *(R5, CRC screening participant, male, age 72, high education)* |
| **Role of emotions/feelings** | Quote 18<br>"Fear of the result [of CRC screening]. I mean, the thought that you might have something, and that you will end up in the medical circuit, with all its consequences. . . colon exams etcetera." *(R18, CRC screening participant, male, age 57, intermediate education)*<br>Quote 19<br>"Well. . . it is like I said: I would like to stay healthy as long as possible and will do anything necessary to make that happen. I want to contribute to that. . . so no, I do not experience any negative emotions with that [CRC screening]. My only thought is, it is good that it is getting done!" *(R9, CRC screening participant, male, 74, high education)* |
| **Role of experience with and views of cancer and preventive screening** | Quote 20<br>"I found that, when you have to deal with it [cancer], it changes the way you think about it. You then truly realise the impact of what it is, and what it does to you." *(R17, CRC screening participant, female, age 73, low education)*<br>Quote 21<br>"Before I got cancer. . . I have that now for. . . it has been five years. . . I believe that this last year I finally started really thinking more about the things I do and do not want." *(R24, CRC screening non-participant, female, age 61, low education)*<br>Quote 22<br>"Well, if it were up to me, I think they should give everyone a yearly scan or something, to screen it all [their body]. But, of course, that is too expensive. But it would be nice if something like that would exist for more types of cancer." *(R3, CRC screening participant, female, age 61, low education)*<br>Quote 23<br>"[Use of preventive screening. . .,] only if there is a reason for it, genetically or familial, things you know." *(R24, CRC screening non-participant, female, age 61, low education)* |
| **A 'good' screening decision** | Quote 24<br>"[A good screening decision is] mainly that it feels right. I believe it is fine to participate in it [CRC screening], so for me that is the right decision. I really do not think about it." *(R12, CRC screening participant, female, age 59, low education)*<br>Quote 25<br>"I just think, they are inviting you, are offering it to you, to get yourself screened. . . so you should seize that opportunity!" *(R6, CRC screening participant, male, age 72, intermediate education)*<br>Quote 26<br>"Because, I believe they offer it [CRC screening] for a reason. I believe prevention is better than cure. I just think it is sort of an obligation to make use of this opportunity. You can only get wiser from it." *(R15, CRC screening participant, female, age 60, low education)*<br>Quote 27<br>"[A good screening decision is] that you stand by it. . . and also that you are well informed. About why you would participate. What the risks are if you do not participate and how you can benefit from it." *(R1, CRC screening participant, female, age 60, high education)*<br>Quote 28<br>"I am very rational, so I think good information is necessary, and then there is also your own feeling. You sometimes feel that you should or should not do something, that is the difficulty with rationality in general, so it is not just reasoning, but also the feeling that comes with that [that makes for a good screening decision]." *(R27, CRC screening non-participant, female, age 72, intermediate education)* |

breast and cervical cancer-screening programmes) has become common and part of daily life. CRC screening non-participants mentioned that it could be difficult to go against this social norm concerning cancer screening. They stated that it did not necessarily strongly influence their personal opinion concerning CRC screening, but they did feel *a* pressure because of it, and a need to explain or defend their decision more (Quote 6, Table 2). This often resulted in them having slight doubts and taking the time to think through their decision once more.

**3.3.2. One's own decision and the involvement of others.**   All CRC screening participants and non-participants experienced the decision whether or not to participate in the national preventive CRC screening programme to be their own decision, as they believed it should be (Quotes 7 and 8, Table 2). Family, friends, or a physician could be consulted when, for example, experiencing doubts, but it remains their own personal decision. Most interviewees did not consult other people concerning their own decision, but they did discuss the CRC screening programme with friends and family as a 'topic of conversation'. A few asked the opinion of their spouse/family, friends with medical knowledge, or current physician concerning cancer or colon problems. In general, it was not considered necessary for the general practitioner to be involved in making the decision concerning CRC screening.

**3.3.3. Role of the government and making screening participation obligatory.**   Several CRC screening participants and non-participants believed that the government had a responsibility in promoting and protecting people's health, but there are boundaries (Quotes 9 and 10, Table 2). Many saw a responsibility for the government to provide preventive cancer screening programmes and public information concerning health and a healthy lifestyle. Some believed that the government should be doing more. However, others believed that the government had no responsibility regarding people's health. Most interviewees saw no active role for the government in the actual decision-making process about participation in the CRC screening programme, apart from providing information. In general, CRC screening participants believed it acceptable to promote CRC screening by making it more known. A few also found advice from the government about participating acceptable. Some CRC screening non-participants found making CRC screening and the national programme more known acceptable too, while others did not. A number of CRC screening non-participants mentioned that the government played a role in creating the social norm and pressure felt regarding screening participation by providing the screening programme and actively inviting people to participate in it. Nearly all CRC screening participants and non-participants believed that CRC screening should not be made obligatory. Some CRC screening participants, however, did express that obligatory participation would be acceptable for them personally, as they thought very highly of CRC screening (Quote 11, Table 2).

**3.3.4. Role of life values.**   Many interviewees viewed participating or not participating in the CRC screening programme directly or indirectly as being associated with what they valued in life. Both CRC screening participants and non-participants valued enjoying life, a good quality of life and spending time with others. Additionally, many CRC screening participants mentioned staying healthy and keeping on living to be important (Quote 12, Table 2). Among CRC screening non-participants several mentioned their health as being important, but more emphasis was put on not worrying too much about what might be, having their freedom, and to live life today, not tomorrow (Quotes 13 and 14, Table 2).

**3.3.5. The perception and personal weighing of the benefits, harms and downsides of CRC screening.**   In general, participants of the CRC screening programme mostly saw possible benefits of CRC screening, where non-participants also, or only, saw possible harms and downsides (Quotes 15 and 16, Table 2). Additionally, for CRC screening participants the possible benefits weighed more heavily than the possible harms and downsides, whereas the reverse was true for CRC screening non-participants. Furthermore, it appears that CRC screening

participants and non-participants differed in their interpretation or explanation of the possible harms, downsides and benefits. For example, CRC screening participants often did not view the probability of a false-positive as a harm or downside, or they saw it as something just inherent to screening (Quote 17, Table 2). CRC screening non-participants, on the other hand, did not always view identifying cancer early on as a benefit as it leads to being labelled as ill/a patient and ending up in medical care, while not feeling ill.

## 3.4. Other aspects involved in the decision-making process

**3.4.1. Role of emotions/feelings.** A sizeable number of CRC screening participants and non-participants indicated that certain emotions/feelings were present (such as a slight fear of cancer or colonoscopy, anxiety while waiting for the test results, anxiety when getting a negative test result, or reassurance when getting a positive test result), and that these played a role in their decision-making process (Quote 18, Table 2). Others, however, stated that this was not the case (Quote 19, Table 2).

**3.4.2. Role of experience with and views of cancer and preventive screening.** Most CRC screening participants and non-participants knew other people close to them who had cancer in the past, and a number of them had cancer in the past themselves. These experiences acted as a source for factual, social and affective information. Both CRC screening participants and non-participants indicated they did not experience a strong fear of cancer, but they did believe it to be a serious illness and thought that the illness and the process of treating it could strongly influence their quality of life. Several interviewees explicitly mentioned that their experiences with cancer influenced what they valued in life and their decision-making process (Quotes 20 and 21, Table 2). Most CRC screening participants were positive about preventive screening in general (Quote 22, Table 2) and participated, or would participate, in other cancer-screening programmes. A few CRC screening participants did not know yet whether they would participate in other cancer-screening programmes. Most CRC screening non-participants explicitly mentioned that their decision to participate in any kind of preventive screening would depend on the disease that was being screened for, the screening procedure and or their personal situation (Quote 23, Table 2).

## 3.5. A 'good' screening decision

Most participants and non-participants of the CRC screening programme mentioned that a 'good' screening decision is one they stand by, that feels right (Quote 24, Table 2), and that is made freely and by themselves. Additionally, all CRC screening non-participants viewed giving their decision substantial thought and making a well-considered decision as important for making a 'good' screening decision, and many also viewed being informed as part of the process. These aspects were mentioned by approximately a quarter of CRC screening participants as being part of making a 'good' screening decision. For half of the CRC screening participants, being offered the opportunity of getting screened for colorectal cancer was mostly what they needed in order to make their decision (Quote 25 and 26, Table 2). Several CRC screening participants and non-participants mentioned that a 'good' screening decision was based on their first impression, often formed using a small amount of information. Most CRC screening participants and non-participants mentioned multiple aspects as being part of making a 'good' screening decision, with often the combination of it being based on reasoning as well as feelings or intuition (Quotes 27 and 28, Table 2). A number of CRC screening participants and non-participants explicitly stated that a 'good' screening decision is also related to the outcome of the decision and its consequences (i.e. not getting cancer or becoming seriously ill). In this light though, some interviewees mentioned that having a different outcome than anticipated does not make

the decision to participate or not participate any less 'good'. The same aspects viewed as being important regarding making a 'good' CRC screening decision were also viewed as being important regarding making a 'good' decision concerning health-related decisions in general.

## 4. Discussion

### 4.1. General results

In our study, both people who decided to participate in the Dutch national preventive CRC screening programme and people who decided not to participate viewed a 'good' screening decision as one they could stand by, that feels right, and that is made freely and by themselves (i.e. referring to autonomy aspects). Although all interviewees experienced that they were free to decide whether they wanted to participate in the CRC screening programme or not, many CRC screening non-participants experienced their decision as not being a completely free decision because they felt a certain social pressure to participate. Another key aspect for making a 'good' screening decision for all interviewees was that it was based on both reasoning and feelings or intuition. A substantial number of interviewees also believed making an informed and deliberative CRC screening decision to be essential. However, in our sample, this view appeared to be more evident among CRC screening non-participants than among participants.

### 4.2. Autonomous and informed decision-making and a social norm

In general, people in our sample made a decision concerning CRC screening participation that they felt was a right decision for them, and in a manner that was right for them. This indicates self-constitution being realised, despite the conditions for it not necessarily being explicitly present within the concept of informed decision-making. However, the social pressure expressed by CRC screening non-participants in our sample to participate in the CRC screening programme may be affecting the freedom of choice of especially those considering not to participate in CRC screening. The notion of there being a positive norm regarding participating in the CRC screening programme is supported by several studies showing a generally positive view towards CRC screening and other types of cancer screening [55–59], as well as the relatively high CRC screening uptake of 73 percent in the Netherlands [33]. This positive social norm regarding participating in cancer screening is not surprising. Firstly, it seems logical that the idea of identifying cancer early on being good simply makes sense to people and would thus appeal strongly to them. Secondly, studies have shown that the government providing or recommending cancer screening and actively inviting people to participate in it, as is the Dutch situation, can signal to people that participating is a good thing to do [40–42]. Lastly, for decades now, the main communication about cancer screening consisted of conveying to people that it was a good thing to do and that they should participate in it [18, 26–28]. As it became more apparent that cancer screening also involves possible harms and downsides [18, 19], many experts in the field of cancer screening found it increasingly important for people to make a personal and informed screening-decision, for which balanced information about both the possible benefits and the harms and downsides concerning cancer screening should be available [8–10]. This development is also related to the societal shift of personal responsibility and autonomous decision-making becoming more prominent [1, 7]. However, the emphasis on cancer-screening participation being one's personal and informed decision is therefore a relatively recent development. To foster freedom of choice and making informed decisions concerning CRC screening participation, the possibility of a previously established social norm should be acknowledged and addressed in public communication regarding CRC screening. It should also be realised that information about CRC screening is likely to be processed within the frame of CRC screening being beneficial. In our study sample, we did not

systematically assess the knowledge levels of our interviewees. However, there is literature suggesting that this existing frame of CRC screening being beneficial could hinder people in acquiring information about the possible harms and downsides of CRC screening, because it could make them inclined to notice and value information about the benefits of CRC screening more, and to interpret the possible harms and downsides, such as false-positives, as not being harms or downsides [60–62].

## 4.3. Informed decision-making and 'good' decision-making in practice

Most CRC screening participants and non-participants in our sample wanted to have some notion of what the CRC screening programme entails and its purpose. However, in our sample, especially CRC screening non-participants appeared to attribute an important role to information and deliberation when deciding about CRC screening participation, similar to the view of cancer-screening experts with their focus on informed decision-making (7–9). The limited number of CRC screening non-participants in our sample with a low education level might have affected this finding. However, among CRC screening participants, it were not only low educated interviewees who did *not* view an informed and deliberative decision as essential for a 'good' decision. It is possible that people who decide not to participate in the CRC screening programme are inclined to seek out more information and to deliberate their decision more than those who decide to participate in it. A contributory factor could be a pressure felt to present sound arguments for not participating in the CRC screening programme as it deviates from the apparent social norm of participating. That being said, we should realise that, in general, our sample is too small to draw definitive conclusions from. The proposed possibility would be, however, interesting to further explore in future research.

Within the concept of informed decision-making, emphasis is put on being fully informed and having sufficient knowledge of all aspects surrounding screening [8, 10, 23, 24]. In our study, however, several CRC screening participants as well as non-participants made their decision based on a first impression using a small amount of information. Furthermore, for half of the CRC screening participants, essentially all that was required to decide to participate was to be invited. This suggests that at least part of the eligible CRC screening population might think differently than cancer-screening experts about the need or interpretation of being fully informed. However, as we did not systematically assess people's actual knowledge about the different aspects of screening, we do not know whether people may have been well informed regardless. An additional finding is that acquired information might not necessarily be used for *making* the screening decision, but more for knowing what to expect once the decision to participate in CRC screening has been made. Supporting this notion, we found that most interviewees had already made their decision before actually receiving the CRC screening programme invitation and official information brochure. This could lead to a biased reading of the brochure, where people are more likely to process information in such a way that it confirms the decision they have already made rather than to form a balanced opinion. As described in the paragraph above, this could contribute to a different awareness, weighing and interpretation of the benefits, harms and downsides of CRC screening [60–62]. Most CRC screening participants and non-participants also used experiences as an information source and found it essential that their decision 'felt right'. These aspects are not necessarily mentioned as part of the concept of informed decision-making. Furthermore, people's values broader than their attitude or test-preferences towards screening were clearly visible in their decision-making process and deliberations. For many people in our sample, their decision regarding CRC screening participation was associated with what they valued in life. CRC screening participants mostly decided to be screened because they valued staying healthy and

their future quality of life. CRC screening non-participants, on the other hand, mostly decided not to be screened because they valued *not* prematurely being labelled as ill/a patient and their current quality of life. Considering our study sample size, we should be careful drawing conclusions, but perhaps our findings indicate a difference in the values and perspective between people deciding to participate in CRC screening and people deciding not to participate. To conclude then, although for most people information and reasoning is part of the decision-making process, the strong emphasis on making a fully informed and well-considered screening decision according to the expert-defined concept of informed decision-making might not be entirely reflecting what the eligible CRC screening population considers to be important for making a 'good' screening decision.

## 4.4. The responsibility of screening providers

In light of fostering autonomous and informed decision-making, providers of CRC screening and other types of cancer screening and/or those who recommend people to participate in it (e.g. the government) have a responsibility to make adequate information concerning it available [8–10]. Balanced, comprehensible and transparent information should be available for those who want to use it. For us, this includes addressing the fact that CRC screening exists because on a population level the benefits outweigh the harms and downsides, but that it is a different matter whether this is also the case on an individual level [9, 18, 20]. Providing the public with balanced, comprehensible and transparent information not only contributes to enabling people to make an informed and well-considered decision concerning CRC screening, but also to establishing a trustworthy provider of cancer screening. This is particularly essential for those mainly participating in the CRC screening programme because they were invited to do so, as that can only be a 'good' decision if the trust people thereby place in the one providing the cancer screening is justified. Additionally, it might be worth considering, or at least starting a debate about, revising or adjusting the active screening invitation strategy that is currently part of the Dutch cancer screening programmes (as well as being used in a number of other countries [29, 30]). An active invitation strategy can have significant positive effects, mainly eliminating certain practical barriers for participating in CRC screening and more equal access to CRC screening (e.g. improved participation rates of people from lower SES groups) [52, 63, 64]. However, using an active invitation strategy may also have an undesired effect. It might be contributing to the perpetuation of the existing social norm regarding participation [40–42] as well as to people possibly not considering what CRC screening participation actually entails [65]. Perhaps the existence of a certain barrier, such as requesting the stool test themselves, is a desired stimulant for people to actively think about their screening decision and consider the possible implications of participating and not participating. Thus, providers of CRC screening are confronted with the complex matter of, on the one hand, enhancing equal access to screening and, on the other hand, promoting informed decision-making and informed participation/non-participation. Therefore, providers should adequately discuss both objectives and all pros and cons associated with a chosen invitation strategy.

## 4.5. Strengths and limitations

Our qualitative study offers relevant insights regarding the concept of informed decision-making in the context of CRC screening and contributes to the debate on how best to support people in making a decision about CRC screening. Our interview sample consisted of a sample of both participants and non-participants of the CRC screening programme, which was divers in sex, age, education, relationship status and having children. Considering our small sample size, the differences in sociodemographic characteristics between CRC screening participants and

non-participants could not be analysed. However, we only interviewed one person with a low education level who did not participate in the CRC screening programme. This group was underrepresented in our 'starting-pool' of possible interviewees to begin with and proved, for diverse reasons, to be more difficult to actually schedule an interview with, which affects generalizability and limits the conclusions we can draw. It is unfortunately a common issue that people from groups with a lower education or social-economic status are more difficult to include in research. Although we stopped conducting interviews when no new main themes were identified in the data, we ideally would have interviewed more CRC screening non-participants with a low education level for a more rounded-out sample and perhaps stronger findings. Additionally, our study did not include people from the age-range 65–69, due to the gradual implementation of the CRC screening programme. Our sample was also diverse in that we included both people who had already been invited to participate in CRC screening and made their decision and people who had not yet been invited to participate but who were aware that they were soon going to be invited. Between the two groups we found no real differences. However, possible limitations are also associated with using either of these groups, as statements made by those who had already acted on their decision might be influenced by screening outcomes. Additionally, statements made by those who did not yet have to act on their decision might be referring to a hypothetical situation, even though they stated they had already made up their mind about participation. Another limitation might be that we used members from a national internet panel as participants. People who participate in online research may differ in significant ways from people who do not participate in online research, affecting generalizability.

In our study, we focused on CRC screening and not also on other types of cancer screening because in the different screening programmes different screening tests are being used. However, the benefits, harms and downsides associated with CRC screening are, in general, similar to the benefits, harms and downsides associated with other types of cancer screening. Additionally, although not the standard situation worldwide, the organisation of CRC screening in the Netherlands is similar to its organisation in a number of other countries, such as the United Kingdom (UK), Australia, Italy and Finland [29, 30]. Therefore, we believe our study results are relevant for other types of cancer screening and other countries.

## 5. Conclusion

In our sample of the eligible CRC screening population, aspects related to the concepts of autonomous and informed decision-making were considered important for making a 'good' decision about participation in the Dutch national preventive CRC screening programme. However, in particular the existence of a social norm may be affecting a true autonomous decision-making process. Additionally, the present, expert-defined, concept of informed decision-making with its strong emphasis on making a fully informed and well-considered screening decision and relatively narrow view concerning values/'self-constitution' does not appear to be entirely reflective of the CRC screening decision-making process in practice. To optimally support people in their decision-making process concerning CRC screening, as well as possibly other cancer-screenings, more efforts could be made to acknowledge the existing social norm regarding participation and to attune to people's diverse values and informational and affective needs regarding their decision-making process.

## Supporting information

**S1 File. Interview guide (in English and Dutch).**
(PDF)

## Author Contributions

**Conceptualization:** Linda N. Douma, Ellen Uiters, Marcel F. Verweij, Danielle R. M. Timmermans.

**Data curation:** Linda N. Douma.

**Formal analysis:** Linda N. Douma, Ellen Uiters.

**Funding acquisition:** Ellen Uiters.

**Investigation:** Linda N. Douma.

**Methodology:** Linda N. Douma, Ellen Uiters, Marcel F. Verweij, Danielle R. M. Timmermans.

**Project administration:** Ellen Uiters.

**Supervision:** Ellen Uiters, Danielle R. M. Timmermans.

**Writing – original draft:** Linda N. Douma.

**Writing – review & editing:** Ellen Uiters, Marcel F. Verweij, Danielle R. M. Timmermans.

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
