## [Decision Letter · Decision Letter 0]

5 Feb 2020

PONE-D-19-36034

Autonomous and informed decision-making in real life: The case of colorectal cancer screening

PLOS ONE

Dear Linda Douma,

Thank you for submitting your manuscript to PLOS ONE. After careful consideration, we feel that it has merit but does not fully meet PLOS ONE’s publication criteria as it currently stands. Therefore, we invite you to submit a revised version of the manuscript that addresses the points raised during the review process.

We would appreciate receiving your revised manuscript by Mar 21 2020 11:59PM. To enhance the reproducibility of your results, we recommend that if applicable you deposit your laboratory protocols in protocols.io, where a protocol can be assigned its own identifier (DOI) such that it can be cited independently in the future. For instructions see: http://journals.plos.org/plosone/s/submission-guidelines#loc-laboratory-protocols

We look forward to receiving your revised manuscript.

Kind regards,

Andrea Gsur

Academic Editor

PLOS ONE

Journal Requirements:

2. Please include additional information regarding the interview guide used in the study and ensure that you have provided sufficient details that others could replicate the analyses. For instance, if you developed a guide as part of this study and it is not under a copyright more restrictive than CC-BY, please include a copy, in both the original language and English, as Supporting Information. In addition, please include further details concerning the development or pre-testing of this guide.

Reviewers' comments:

Reviewer's Responses to Questions

**Comments to the Author**

1. Is the manuscript technically sound, and do the data support the conclusions?

Reviewer #1: Partly

Reviewer #2: Yes

Reviewer #3: Partly

2. Has the statistical analysis been performed appropriately and rigorously? 

Reviewer #1: N/A

Reviewer #2: Yes

Reviewer #3: N/A

3. Have the authors made all data underlying the findings in their manuscript fully available?

Reviewer #1: Yes

Reviewer #2: Yes

Reviewer #3: No

4. Is the manuscript presented in an intelligible fashion and written in standard English?

Reviewer #1: Yes

Reviewer #2: Yes

Reviewer #3: Yes

5. Review Comments to the Author

Reviewer #1: The authors are presenting the results of a survey of subjects eligible for CRC screening, who have been interviewed, using in-depth interviews, about their experience of decision making within the Dutch CRC screening program.

The questionnaire was designed to assess the weight of the different components of an autonomous decision, referring to a comprehensive conceptual framework, considering the different components involved in the definition of individual’s autonomy.

The adoption of such framework allows the authors to analyze factors influencing subjects decision to participate in screening, going beyond the usual concept of informed decision making, identifying issues related to social norms as well as to individual’s values and experiences which can play a role in the decision making process, above and beyond the level of knowledge about screening and CRC.

This represents therefore a valuable contribution to the debate focused on the need to promote informed decision making in screening.

The paper is however quite long. Although the description of the framework is certainly useful for the readers, the authors, when presenting the results, might focus more on those aspects which can specifically be derived using a qualitative approach, while other aspects, such as, for example, the evaluation of the information provided by the official brochure (several studies have already provided data about the perceived quality of screening leaflets and the need for information), might be shortened.

The number of subjects interviewed is really small, and as indicated also by the authors, subjects in the lowest educational level are severely under-represented in the group of non-participants.

I am not an expert in the field of qualitative research and I can imagine that the process of fixing, conducting and analysing these interviews is resource demanding. So the choice to stop the interviews when no new themes emerged, might be justified both from a methodological point of view as well as from a feasibility perspective. However, I would think that this choice may allow to indicate which are the themes reported by the different subgroups, but it may not allow to conclude whether a specific theme is associated with different sub-groups (participants or non-participants, for example), or which is the its relative weight as compared to other factors. I would consider to address this issue in the discussion.

The final suggestion to consider the option to reconsider the choice of adopting an active invitation strategy, based on the reported feeling of social pressure to participate by non- participants. The unbalanced distribution of the strata of educational level and the limited size do not allow to reach sound conclusion about the weight of such factor. It definitely plays a role (as also reported from other studies) and the authors are also describing the potential influence of such social norm on the way people are processing information about screening. On the other hand, the invitation does not only represent a strategy aimed to overcome practical barrier. Strong evidence is indeed supporting the hypothesis that an active invitation strategy is associated with a reduction of disparities in access to screening (and subsequent caner care). Self-initiated screening is more often a decision of better educated people, showing higher health awareness and the necessary competence, knowledge and literacy skills to consider taking screening. Avoiding widening the SES gap while promoting informed choice represents a challenge which should not be underevaluated

Reviewer #2: In the manuscript the authors are presenting very interesting and still understudied field of autonomous and informed decision-making in cancer screening – precisely in colorectal cancer screening. Therefore, I find the study results of high importance for health authorities and personnel that are involved in the cancer screening and its organization.

Certain parts of the manuscript are difficult to understand (for instance lines 27, 28-29, 214), I would advise that a native speaker revises the text.

I would advise also to consider using the word “harms” instead of “downsides”.

Lines 251-256 should go into discussion part.

Reviewer #3: Thanks for the opportunity to review this paper. I provide both general and specific comments below:

GENERAL COMMENTS

1. Rationale for study:

- Rationale for focus on autonomy in addition to informed decision is set out as a key rationale for the paper. But it is not clear if CRC screening conducted through primary care providers would not meet the shared decision-making context, and how autonomy might be supported in that context.

- The rationale for focusing solely on CRC screening for understanding something more generic – autonomous and informed decision-making – was not clearly laid out. Is there anything about cancer screening decisions or CRC screening decisions in particular that may affect the study results? The authors cite three articles for why they focus on CRC screening but don’t explain the rationale in the paper. They need to make a better case for why they focus on CRC screening and what the implications of that are.

- It seems that these findings have fairly narrow relevance to the case of CRC screening. Even generalizing to other types of cancer screening may be problematic given the unique nature of the screening test, follow-up of positive tests, etc.

2. ‘Good’ decision

- What is a ‘good’ decision? Decision conflict? Decision confidence? Decision satisfaction? Not clear how the authors define what is a ‘good’ decision, despite using quotations for ‘good’ throughout the paper. Although the a ‘good’ decision is the focus on the second research question, the results are not clearly presented. Were participants asked if they felt they had made a ‘good’ decision, or only to comment on what they thought a ‘good’ decision was? This needs to be clarified. If all participants indicate that they made a good decision, then they are really justifying how they made their decision – which is a different consideration.

- The contrast between informed and shared decision-making is briefly noted, but this deserves more attention in the paper. Is this the case because invitations to CRC screening in NL are through letter invitation and not through visit with a primary care provider?

3. Methodology:

- The methodology/study design is not explained. The authors note that in depth interviews will be used to collect data but no overarching study design is noted.

- Unclear why study relied solely on interviews. Further to above point, a case study approach of different programs would have strengthened the study considerably and allowed for more generalizable results.

- Important limitations of studying decision-making with only one programme context (NL). Why a broader study in other jursidctions wasn’t contemplated is unclear, but would have led to a much stronger study.

- Used national online research panel – unclear why given the numbers (only 27 interviewees), a more traditional (and less biased – they have to sign up and they receive a 20 EUR gift card) approach to recruit participants wasn’t used.

- The two groups were not what I anticipated. The first group were invitees (first time). Second group were not invited. I was expecting the second group who had been invited but decided not to go forward with screening.

- Not clear enough what the sample was. What they had to answer to be eligible.

- The authors make note about challenges recruiting lower education participants (but not clear why scheduling issues were not surmountable).

- Interviewees were asked about experience making CRC screening decision, yet some participants come from a group that had not yet been invited – so what did that entail?

- Why did the authors include those who had already made the decision and those that had not (intention group)? They suggest that this is a strength of the study, but then they combine the results for these two groups, which does not allow the distinction between these two groups to be assessed. This is an important limitation.

- Assume quotes were translated from Dutch to English – how translation was done should be clarified in the methods.

4. Results:

- The characteristics of the two groups differ in terms of sex, children and age (to varying extents). The authors need to unpack the implication of this more.

- Regarding participants, also not clear if any of the sample would be classified as higher risk populations in terms of CRC screening (e.g., have direct family members who had colorectal cancer). The assumption is that the participants were all in the average risk population – but section 3.4.2 possibly suggests otherwise. This should be explicit.

- I appreciated Table 2 – this was an effective way to present qualitative interview data that doesn’t negatively affect the flow of the narrative of the paper.

5. Summary

Overall, while an interesting study, there are many limitations that affect the generalizability of the findings. Part of this is due to the methodological approach, which is quite basic, part of this is conceptual with issues related to the rationale for the study and in particular how ‘good’ decisions were presented both conceptually and empirically. Ultimately, the results are relevant only to CRC screening decisions in the context of public programs that use a mass invitation protocol (rather than a primary care model). This is fairly narrow for an interest in informed/autonomous decision-making – and the authors need to make a much stronger case for how to position these findings in a broader context, or alternatively, make case for why the narrow CRC screening focus is justified.

SPECIFIC COMMENTS:

- The authors use ‘…cancer screenings…’ in a number of places. This should be corrected to ‘…types of cancer screening…’ or ‘…cancer screening programs…’.

- Line 122 – “This process could stimulate…”. It’s not clear whether ‘this’ refers the to sentence immediately preceding it or the sentence before that. This should be tweaked to be clear.

- End of section 2.3 – authors identify emerging themes. This should not be presented in the data collection/analysis section – as this is a methods section. It should be in the results.

- Table 1 should break down the participants by the four groups – not just two amalgamated groups.

- Line 385 – change ‘Oher’ to ‘Other’.

- Line 390 – avoid ‘—,’ together. Either use ‘—’ or ‘,’ only

- Line 394 and 395 – avoid ‘had had’.

6. PLOS authors have the option to publish the peer review history of their article (what does this mean?). If published, this will include your full peer review and any attached files.

Reviewer #1: No

Reviewer #2: No

Reviewer #3: No

---

## [Author Response · Author response to Decision Letter 0]

18 Mar 2020

We thank the reviewers for their comments. See the uploaded document for our response.

---

## [Decision Letter · Decision Letter 1]

4 May 2020

Autonomous and informed decision-making: The case of colorectal cancer screening

PONE-D-19-36034R1

Dear Ms Douma,

We are pleased to inform you that your manuscript has been judged scientifically suitable for publication and will be formally accepted for publication once it complies with all outstanding technical requirements.

With kind regards,

Ker-Kan Tan

Academic Editor

PLOS ONE

Additional Editor Comments (optional):

Reviewers' comments:

Reviewer's Responses to Questions

**Comments to the Author**

1. If the authors have adequately addressed your comments raised in a previous round of review and you feel that this manuscript is now acceptable for publication, you may indicate that here to bypass the “Comments to the Author” section, enter your conflict of interest statement in the “Confidential to Editor” section, and submit your "Accept" recommendation.

Reviewer #2: All comments have been addressed

Reviewer #3: All comments have been addressed

2. Is the manuscript technically sound, and do the data support the conclusions?

Reviewer #2: Yes

Reviewer #3: (No Response)

3. Has the statistical analysis been performed appropriately and rigorously? 

Reviewer #2: Yes

Reviewer #3: (No Response)

4. Have the authors made all data underlying the findings in their manuscript fully available?

Reviewer #2: Yes

Reviewer #3: (No Response)

5. Is the manuscript presented in an intelligible fashion and written in standard English?

Reviewer #2: Yes

Reviewer #3: (No Response)

6. Review Comments to the Author

Reviewer #2: In the manuscript the authors are presenting very interesting and still understudied field of

autonomous and informed decision-making in cancer screening – precisely in colorectal cancer

screening. Therefore, I find the study results of high importance for health authorities and personnel

that are involved in the cancer screening and its organization.

All comments have been addressed. The manuscript is now technically sound, and the data support the conclusions. Statistical analysis has been performed appropriately. The authors made all necessary data underlying the findings available. The manuscript is written in standard English.

Reviewer #3: (No Response)

7. PLOS authors have the option to publish the peer review history of their article (what does this mean?). If published, this will include your full peer review and any attached files.

Reviewer #2: No

Reviewer #3: No

---

## [Editor Report · Acceptance letter]

14 May 2020

PONE-D-19-36034R1 

Autonomous and informed decision-making: The case of colorectal cancer screening 

Dear Dr. Douma:

I am pleased to inform you that your manuscript has been deemed suitable for publication in PLOS ONE. Congratulations! Your manuscript is now with our production department. 

With kind regards,

on behalf of

Dr. Ker-Kan Tan 

Academic Editor

PLOS ONE